# SegFit: Robust SMPL-X Fitting with Body Part Segmentation on Real-World Point Clouds

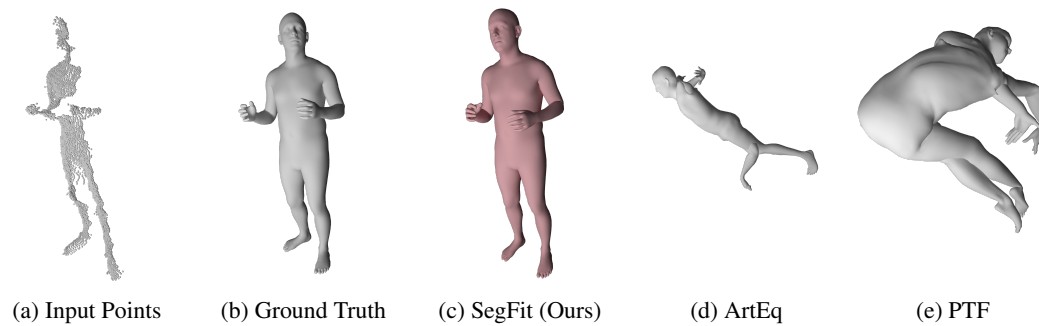

(a) Input Points     (b) Ground Truth     (c) SegFit (Ours)     (d) ArtEq     (e) PTF

Figure 1: Our proposed method SegFit reconstructs human poses from point clouds using body part segmentation and the SMPL-X model (Pavlakos et al., 2019). We showcase SMPL-X fitting results on the EgoBody dataset (Zhang et al., 2022), comparing them to state-of-the-art methods, ArtEq (Feng et al., 2023b) and PTF (Wang et al., 2021b).

## Abstract

Fitting parametric human body models to 3D point cloud is crucial for applications such as virtual reality and human-robot interaction but remains challenging due to the lack of contextual guidance, often leading to imprecise results. To address this, we propose a hybrid approach that incorporates body part segmentation into the fitting process, enhancing pose estimation and segmentation accuracy. Our method starts with an initial segmentation, assigning each point to a specific body part. This segmentation guides a two-step optimization in fitting an SMPL-X model: first, approximating the initial pose and orientation using body part centroids, and second, refining the model by considering the entire point cloud. After fitting, we reassign body parts to the point cloud through nearest-neighbor matching, resulting in more accurate segmentation. This enhanced segmentation serves as pseudo ground truth to fine-tune the segmentation network in a self-supervised manner, creating a feedback loop where improvements in pose fitting lead to better segmentation and vice versa. We evaluate our approach on four challenging datasets – PosePrior, EgoBody, BEHAVE, and Hi4D – demonstrating significant improvements over leading methods, including a tenfold increase in pose modeling accuracy and a 15% enhancement in segmentation accuracy after fine-tuning. Our contributions are twofold: (1) introducing a novel hybrid method that unifies pose fitting and body part segmentation on point clouds, enabling mutual enhancement through iterative refinement; and (2) developing a self-supervised technique for fine-tuning segmentation networks using pseudo ground truths derived from fitted models. This work advances the state of the art in human body fitting to point clouds, facilitating more accurate human representations in complex environments and benefiting applications that require precise human modeling. We will make the source code publicly available.

# 1 INTRODUCTION

Capturing the intricate details of human movement and shape is fundamental to a wide range of applications, from creating realistic avatars in virtual reality (Thalmann & Musse, 2012; Slater & Sanchez-Vives, 2016) to enhancing human-robot interactions (Argall et al., 2009; Billard & Kragic, 2019). Fitting a parametric human body model to 3D point cloud data (measured by, e.g., Lidar or Kinect) without contextual guidance can be challenging and often yields imprecise results (Bogo et al., 2016; Kolotouros et al., 2019). However, having prior information about body parts can significantly improve this process (Varol et al., 2017; Zanfir et al., 2018; Xu et al., 2020). Thus, this paper focuses on incorporating body part segmentation into the body fitting process.

In recent years, parametric models like SMPL-X (Loper et al., 2015) have become instrumental in representing 3D human poses and shapes. These models are typically fitted to point cloud data obtained from depth sensors or multi-view stereo systems using optimization techniques such as gradient descent (Bogo et al., 2016; Pavlakos et al., 2019) or neural networks (Kolotouros et al., 2019; Kocabas et al., 2020). Simultaneously, advancements in body part segmentation —- assigning each point in a cloud to a specific body part —- have provided valuable context for understanding human poses and interactions in applications like computer graphics, personalized healthcare, and autonomous systems (Li et al., 2018; Xu et al., 2020; Takmaz et al., 2023).

Despite these advancements, current methods often struggle when applied to real-world data featuring complex poses, occlusions, multiple interacting individuals, or human-object interactions (Trumble et al., 2018; Moon et al., 2019; Hassan et al., 2019a). This is largely due to existing approaches being trained on simplified datasets with isolated bodies in controlled environments (Mahmood et al., 2019; Andriluka et al., 2014). Consequently, their performance diminishes when confronted with the diversity and unpredictability of real-world scenarios (Mehta et al., 2017; Dabral et al., 2018).

To address these limitations, we propose a hybrid approach that unifies human pose fitting and body part segmentation on point clouds, enabling each process to iteratively inform and enhance the other. Our method begins with an initial segmentation of the point cloud using a state-of-the-art network, Human3D (Takmaz et al., 2023), which provides a preliminary assignment of 3D points to specific body parts. This segmentation acts as a guide for the fitting of the SMPL-X model through a two-step optimization process (Feng et al., 2023b).

The first optimization step uses the centroids of the body parts to approximate the initial pose and orientation of the model (Bogo et al., 2016; Xiang et al., 2019). This step is crucial since it provides a strong starting point for the model parameters, similar to assembling the edge pieces of a puzzle first. The second step refines the model by considering the entire point cloud, adjusting the pose and shape parameters to achieve a more precise fit (Pavlakos et al., 2019; Kolotouros et al., 2019). This refinement fills in the puzzle's interior. After fitting the model, we leverage the SMPL-X mesh to reassign body parts to the point cloud through nearest-neighbor matching (Ge et al., 2019; Remelli et al., 2020). This results in a more accurate segmentation than the initial prediction, effectively updating our "puzzle sketch" based on the assembled pieces. We then use this enhanced segmentation as a pseudo ground truth to fine-tune the segmentation network. This self-supervised learning approach allows the network to improve its performance on new, unlabeled data, such as point clouds derived from in-the-wild captures (Varol et al., 2017; Zhang et al., 2021).

This reciprocal process creates a feedback loop where improvements in pose fitting lead to better segmentation, which in turn facilitates more accurate fitting in subsequent iterations (Zanfir et al., 2018; Kocabas et al., 2020). By integrating these two tasks, we address the challenges of generalizing to diverse datasets and enhance the robustness of both processes in real-world conditions.

We evaluate our approach on four challenging datasets: PosePrior (Zhang et al., 2021), EgoBody (Zhang et al., 2022), BEHAVE (Bhatnagar et al., 2022), and Hi4D (Yin et al., 2023). These datasets encompass a wide range of complex scenarios, including intricate poses, partial occlusions, close human interactions, and human-object interactions (Hassan et al., 2019a; Xiang et al., 2019; Pumarola et al., 2021). Our experiments demonstrate significant improvements in both pose modeling accuracy and body part segmentation performance compared to leading methods (Kolotouros et al., 2019; Kocabas et al., 2020; Zanfir et al., 2018), especially in the complicated scenes. Specifically, we observe a tenfold average improvement in pose modeling accuracy and a 15% en-

hancement in segmentation accuracy after fine-tuning with our pseudo ground truths, as evaluated on EgoBody (Zhang et al., 2022) and BEHAVE (Bhatnagar et al., 2022).

In summary, our contributions are twofold:

1. **A Hybrid Approach Integrating Pose Fitting and Segmentation**: We present a novel method that combines human pose fitting with body part segmentation on point clouds, enabling mutual enhancement through iterative refinement.

2. **Self-Supervised Fine-Tuning of Part Segmentation**: By using the fitted SMPL-X meshes to generate pseudo ground truth (Zanfir et al., 2018; Kocabas et al., 2020; Zhang et al., 2021), we enable the fine-tuning of segmentation networks on unlabeled data, enhancing their performance on diverse real-world datasets.

Our work advances the state of the art in human body fitting to point clouds and has significant implications for applications requiring accurate human representations in complex environments (Thalmann & Musse, 2012; Slater & Sanchez-Vives, 2016; Billard & Kragic, 2019).

## 2 RELATED WORK

Estimating human body pose and shape from 3D point clouds is vital in computer vision, with applications in virtual reality, animation, and human-computer interaction. While extensive research has been conducted on fitting parametric human models to 2D images (Bogo et al., 2016; Kanazawa et al., 2018; Kolotouros et al., 2019), we focus on methods that directly operate on 3D point cloud data. Point clouds capture detailed geometric information and avoid the ambiguities inherent in 2D projections, making them valuable for precise human modeling.

**Methods for Fitting Human Poses onto Point Clouds.** Several approaches have been developed to fit human poses to point clouds. Bhatnagar et al. (2020) introduced *IP-Net*, which combines implicit representations with parametric models to reconstruct clothed human bodies from partial scans. IP-Net learns a continuous occupancy field representing the human body, allowing for detailed reconstructions even with incomplete data. Wang et al. (2021b) proposed *PTF*, a method that fits SMPL models to point clouds by considering local geometric features. By leveraging local point distributions, PTF improves fitting accuracy in areas with high curvature or fine-grained details. Zuo et al. (2021) presented a self-supervised approach for 3D human motion reconstruction from depth sequences. Their method leverages temporal coherence without requiring ground truth annotations, effectively reconstructing dynamic human motions. Cai et al. (2023) developed *PointHPS*, a hierarchical point-based network that directly regresses SMPL parameters from point clouds. PointHPS achieves state-of-the-art results by utilizing a point-based encoder-decoder architecture.

The common challenges among these methods include sensitivity to initialization, reliance on large annotated datasets, and difficulties in handling complex poses, occlusions, and interactions. Regression-based methods often require extensive training data to cover the variability in human shapes and poses, which may not be feasible for all applications. Additionally, they may not effectively capture subtle variations in body shape or handle occlusions caused by interactions with objects or other people.

Our approach differs by integrating body part segmentation into the fitting process, leveraging initial segmentation to improve model initialization and fitting accuracy. By using body part information, we can distinguish between symmetric limbs and reduce ambiguities in global orientation, which is a common issue in optimization-based methods. This integration allows us to achieve robust initialization without the need for multiple optimization runs from different starting points, as done in methods like SMPLify-X (Pavlakos et al., 2019). Furthermore, our method improves the fitting accuracy by focusing on both global and local alignment between the model and the point cloud. The initial segmentation guides the fitting process, making it more resilient to noise and outliers. By considering the centroids of body parts during initialization and refining the fit using the entire point cloud, we enhance the ability of the model to capture complex poses and interactions.

**Datasets and the Challenge of Annotations.** High-quality datasets with accurate annotations are crucial for training and evaluating human body modeling methods. However, obtaining datasets with detailed body part segmentation on point clouds is particularly challenging. Existing datasets

like Hi4D (Yin et al., 2023), BEHAVE (Bhatnagar et al., 2022), and EgoBody (Zhang et al., 2022) provide valuable data capturing complex human poses, interactions, and motions. Hi4D offers high-fidelity 4D data for human reconstruction in motion, BEHAVE focuses on human-object interactions, and EgoBody captures human activities from an egocentric perspective. Despite their richness, these datasets often lack detailed body part segmentation annotations necessary for training segmentation networks. Annotating body parts in 3D point clouds is labor-intensive and time-consuming, making it impractical for large datasets. This scarcity of annotated data limits the effectiveness of supervised learning methods for body part segmentation on point clouds.

Our method addresses this challenge by generating improved body part segmentations through the fitted SMPL-X models. After fitting the model to the point cloud, we use the correspondences between the model vertices and the point cloud to assign body part labels, effectively creating pseudo ground truth annotations. This process enhances the initial segmentation obtained from networks like Human3D (Takmaz et al., 2023). By utilizing these improved segmentations, we can fine-tune segmentation networks in a self-supervised manner. This approach does not require additional manual annotations and enables the network to generalize better to diverse and complex datasets. Our method contributes to the enrichment of existing datasets with high-quality segmentation labels, facilitating better training data for future research and applications.

## 3 METHOD

This section presents our method for fitting the SMPL-X parametric human body model to 3D point clouds by leveraging initial body part segmentation. Our approach consists of a two-step process: an initialization phase that provides a reliable starting point for optimization and a fitting phase that refines the model parameters to represent the human subjects in the point cloud accurately. We further utilize the fitted models to enhance body part segmentation through a self-supervised fine-tuning process.

### 3.1 PROBLEM DEFINITION

Given a 3D point cloud $\mathcal{P} = \{\mathbf{p}_i\}_{i=1}^{N}$ capturing one or more human subjects, our goal is to estimate the pose and shape parameters of the SMPL-X model (Pavlakos et al., 2019), resulting in accurate 3D meshes that align with the point cloud. The challenge lies in initializing the model parameters to avoid local minima during optimization and handling ambiguities due to the symmetry of the human body. To address this, we propose a method that leverages initial body part segmentation obtained from Human3D (Takmaz et al., 2023). This segmentation provides valuable information that aids in distinguishing between symmetric body parts and improves the initialization of the SMPL-X model.

### 3.2 OVERVIEW OF THE APPROACH

Our method comprises the following main components:

1. **Initial Body Part Segmentation**: Segment the input point cloud into body parts using Human3D (Takmaz et al., 2023), assigning each point a body part label.

2. **Model Initialization**: Compute centroids of the segmented body parts and use them to initialize the SMPL-X model parameters, including pose, shape, and global orientation.

3. **Model Fitting**: Refine the SMPL-X model by optimizing an objective function that balances data fidelity and regularization terms, fitting the model to the entire point cloud.

4. **Improved Body Part Segmentation**: Use the fitted SMPL-X meshes to reassign part labels to the point cloud via nearest-neighbor matching, resulting in enhanced segmentation.

5. **Self-Supervised Fine-Tuning**: Fine-tune the Human3D segmentation network using the improved segmentation as pseudo ground truth, enhancing its performance on new data.

We provide detailed descriptions of each component in the following subsections.

### 3.3 INITIAL BODY PART SEGMENTATION

We begin by applying Human3D (Takmaz et al., 2023), a state-of-the-art body part segmentation network, to the input point cloud $\mathcal{P}$. This network assigns a body part label $s_i \in \{1, \ldots, K\}$ to each point $\mathbf{p}_i$, where $K$ is the total number of body parts defined by the SMPL-X model. The segmentation provides an initial understanding of the spatial arrangement of different body parts within the measured 3D point cloud.

### 3.4 MODEL INITIALIZATION

Accurate initialization is crucial for effective model fitting, as poor initialization can lead to suboptimal solutions or cause the optimizer to become trapped in local minima. Traditional methods like SMPLify-X (Pavlakos et al., 2019) and PROX (Hassan et al., 2019b) handle orientation ambiguities by fitting the model multiple times with different initial rotations. In contrast, we leverage body part segmentation to obtain a reliable initialization in a single attempt.

For each body part $k \in \{1, \ldots, K\}$, we compute the centroid of the corresponding points in the point cloud as follows:

$$\mathbf{c}_k^{\text{scan}} = \frac{1}{N_k} \sum_{i=1}^{N_k} \mathbf{p}_i, \tag{1}$$

where $N_k$ is the number of points assigned to body part $k$. The same is done for the body parts in the SMPL-X template model $\mathcal{M}_0$.

We estimate an initial global rotation $\mathbf{R}_0$ and translation $\mathbf{t}_0$ and create a rough approximation of the pose by running the same fitting process as described in Section 3.5, however, considering only the described centroids and neglecting the remainder of the point cloud. This alignment provides a fast and reliable initialization for the global pose and reduces ambiguities in orientation by utilizing the spatial distribution of body parts.

### 3.5 MODEL FITTING

With the initialized model parameters, we refine the SMPL-X model by fitting it to the entire point cloud. The SMPL-X model (Pavlakos et al., 2019) is parameterized by pose $\boldsymbol{\theta} \in \mathbb{R}^{J \times 3}$, shape $\boldsymbol{\beta} \in \mathbb{R}^B$, and global translation $\mathbf{t} \in \mathbb{R}^3$, where $J$ is the number of joints and $B$ is the number of shape coefficients. Our optimization aims to find the model parameters that minimize the following objective function:

$$\mathcal{L} = \lambda_{\text{data}} \mathcal{L}_{\text{data}} + \lambda_{\text{pose}} \mathcal{L}_{\text{pose}} + \lambda_{\text{shape}} \mathcal{L}_{\text{shape}}, \tag{2}$$

where $\lambda_{\text{data}}$, $\lambda_{\text{pose}}$ and $\lambda_{\text{shape}}$, are weighting factors balancing the contributions of each term.

**Data Term.** The data term measures the discrepancy between the point cloud and the model surface. We use a robust one-sided Chamfer distance based on the Huber loss to reduce sensitivity to outliers as follows:

$$\mathcal{L}_{\text{data}} = \sum_{k=1}^{K} \sum_{i=1}^{N_k} \min_{\mathbf{v} \in \mathcal{V}_k} \mathcal{L}_{\text{Huber}}(\mathbf{p}_i - \mathbf{v}), \tag{3}$$

where $\mathcal{V}k$ is the set of vertices corresponding to body part $k$, and $\mathcal{L}_{\text{Huber}}$ is defined as:

$$\mathcal{L}_{\text{Huber}}(\mathbf{r}) = \begin{cases} \frac{1}{2} |\mathbf{r}|_2^2 & \text{if } |\mathbf{r}|_2 \leq \delta, \\ \delta(|\mathbf{r}|_2 - \frac{1}{2}\delta) & \text{otherwise,} \end{cases} \tag{4}$$

with $\delta$ being the Huber loss parameter. This formulation ensures that large residuals are penalized linearly, reducing the influence of outliers.

To handle high-resolution scans and manage computational complexity, we limit the number of points considered in the data term by randomly subsampling $n$ points from each body part.

**Pose and Shape Regularization.** We include regularization terms on the pose and shape parameters to encourage plausible human poses and shapes as follows:

$$\mathcal{L}_{\text{pose}} = |\boldsymbol{\theta} - \boldsymbol{\theta}_0|_2^2, \quad \mathcal{L}_{\text{shape}} = |\boldsymbol{\beta}|_2^2, \tag{5}$$

where $\boldsymbol{\theta}_0$ is the initial pose from the model initialization.

To further enforce realistic poses and avoid unnatural articulations, we utilize VPoser, a variational human pose prior trained on a large dataset of human poses. VPoser encodes the pose parameters into a low-dimensional latent space, modeling the distribution of natural human poses.

**Optimization.** We optimize the objective function $\mathcal{L}$ with respect to the model parameters $\boldsymbol{\theta}$, $\boldsymbol{\beta}$, $\mathbf{R}$, and $\mathbf{t}$ using the Adam optimizer, which adapts the learning rates for each parameter and is suitable for large-scale optimization.

### 3.6 IMPROVING BODY PART SEGMENTATION

After obtaining the fitted SMPL-X model, we enhance the body part segmentation of the point cloud by reassigning labels based on the fitted model. For each point $\mathbf{p}_i$, we find its nearest vertex $\mathbf{v}_i^*$ on the SMPL-X mesh:

$$\mathbf{v}i^* = \arg \min_{\mathbf{v} \in \mathcal{V}} |\mathbf{p}_i - \mathbf{v}|_2. \tag{6}$$

We then assign the body part label of $\mathbf{v}_i^*$ to $\mathbf{p}_i$:

$$s_i = s(\mathbf{v}_i), \tag{7}$$

where $s(\mathbf{v}_i)$ denotes the body part label of vertex $\mathbf{v}_i$ in the SMPL-X model.

This process results in a refined segmentation $\mathcal{S}^* = \{s_i\}_{i=1}^N$ that is more accurate than the initial segmentation from Human3D. The improved segmentation benefits from the fitted model adherence to the point cloud and the SMPL-X model.

### 3.7 SELF-SUPERVISED FINE-TUNING

The refined segmentation $\mathcal{S}^*$ serves as pseudo ground truth for self-supervised fine-tuning of the Human3D segmentation network. By training the network on these improved labels, we enable it to generalize better to new, unlabeled data and enhance its segmentation accuracy in diverse scenarios.

The fine-tuning process involves minimizing the cross-entropy loss between the network predictions and the pseudo ground truth labels:

$$\mathcal{L}_{\text{seg}} = -\sum_{i=1}^N \sum_{k=1}^K y_{i,k} \log p_{i,k}, \tag{8}$$

where $y_{i,k} = 1$ if $s_i^* = k$ and 0 otherwise, and $p_{i,k}$ is the predicted probability that point $\mathbf{p}_i$ belongs to body part $k$. This self-supervised learning approach creates a feedback loop where improved model fitting leads to better segmentation, which in turn facilitates more accurate model fitting in subsequent iterations.

### 3.8 SUMMARY

Our method effectively combines body part segmentation and model fitting to enhance the accuracy of both processes. By initializing the SMPL-X model using body part centroids and refining it through optimization, we achieve accurate alignment with the input point cloud. The fitted models then improve segmentation, which can be used to fine-tune the segmentation network in a self-supervised manner. This iterative process addresses challenges in pose estimation and segmentation, particularly in complex real-world scenarios.

## 4 EXPERIMENTS

We evaluate our proposed method on several challenging datasets to demonstrate its effectiveness in modeling human poses and improving body part segmentation from point clouds. Our experiments are designed to assess the accuracy and efficiency of our method compared to baseline approaches, as well as to highlight the improvements in segmentation accuracy achieved through our self-supervised fine-tuning process.

We evaluate our method through a series of steps and compare the results with the baseline models, ArtEq (Feng et al., 2023b) and Human3D (Takmaz et al., 2023). First, we analyze the performance of our pose modeling approach across various conditions and configurations, benchmarking it against ArtEq. Next, we examine the improvements in body part segmentation facilitated by the pose modeling, comparing these to the initial segmentation accuracy of Human3D. Finally, we assess the impact of fine-tuning Human3D using the enhanced part segmentations as pseudo-ground truths. Our evaluations are conducted on four distinct datasets, detailed below.

To thoroughly evaluate the generalization capabilities of our method, we conduct experiments on four diverse datasets, each presenting unique challenges:

1. **PosePrior** (Zhang et al., 2021): A subset of the AMASS dataset containing complex human poses with limbs extended at extreme angles, posing difficulties for accurate pose estimation.

2. **BEHAVE** (Bhatnagar et al., 2022): Consists of dense scans of humans interacting closely with objects. Each point cloud includes a single human and one object, challenging the method to distinguish between human and object and to handle occlusions.

3. **EgoBody** (Zhang et al., 2022): Captures human activities from an egocentric perspective, featuring scenes with two interacting humans and background elements that need to be separated during segmentation.

4. **Hi4D** (Yin et al., 2023): Features high-resolution scans of humans engaged in close interactions, such as fighting or dancing, requiring the method to identify boundaries between individuals accurately.

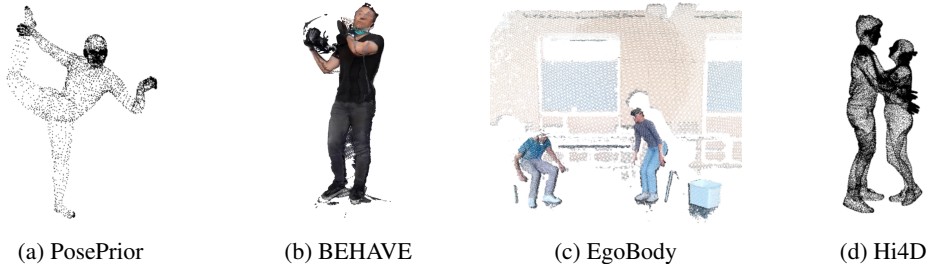

(a) PosePrior       (b) BEHAVE       (c) EgoBody       (d) Hi4D

Figure 2: Examples of point clouds from the datasets used in this study, illustrating variations in pose, occlusions, and interactions present in the PosePrior (AMASS), BEHAVE, EgoBody, and Hi4D datasets.

**Evaluation Metrics**: Following prior work, we evaluate the accuracy of the fitted models by measuring the Euclidean distance in millimeters between corresponding vertices (V2V error) and joints (J2J error) of the fitted and ground-truth SMPL-X models. We also record the average processing time taken to fit a single human instance.

### 4.1 POSE ESTIMATION RESULTS

We compare our method, referred to as *SegFit*, with the state-of-the-art PTF (Wang et al., 2021b) and ArtEq (Zhang et al., 2021). We do not consider other baselines, such as IP-Net (Bhatnagar et al., 2020) and LoopReg (Kolotouros et al., 2019), since they lead to less accurate results than PTF and ArtEq.

To enable the comparison, it is necessary to adjust the inputs to PTF and ArtEq as these methods are not designed to handle multi-human scenes. Hence, before evaluating them on EgoBody and Hi4D, we first separate the human instances based on ground truths and pass the corresponding point clouds into PTF and ArtEq individually. This also includes removing the background in EgoBody. Furthermore, as Human3D struggles to differentiate between touching instances in Hi4D, we finetune on this dataset before evaluating SegFit. The resulting instance segmentation is then used as a basis for SegFit and is at most as precise as the ground truth separation we perform for PTF and ArtEq.

From Table 1, we observe that the proposed SegFit significantly outperforms ArtEq and PTF on BEHAVE, EgoBody, and Hi4D datasets, demonstrating better generalization to diverse and complex real-world scenarios. On the BEHAVE dataset, SegFit achieves a V2V error six times lower than ArtEq, indicating superior performance in handling human-object interactions. On the EgoBody dataset, SegFit reduces the V2V error by an order of magnitude compared to ArtEq and PTF, showcasing robustness in scenes with multiple humans and background clutter.

Although ArtEq slightly outperforms SegFit on the PosePrior dataset, SegFit still achieves competitive results. Additionally, SegFit provides a good balance between accuracy and computational efficiency, with fitting times significantly lower than PTF and only slightly higher than ArtEq.

| Dataset | Metric | PTF (Wang et al., 2021a) | ArtEq (Feng et al., 2023a) | SegFit (Ours) |
|---|---|---|---|---|
| BEHAVE | V2V [mm] | 209.3 | 246.0 | **37.0** |
| | J2J [mm] | 245.9 | 279.4 | **30.7** |
| | Time [s] | 30.771 | **0.102** | 1.860 |
| EgoBody | V2V [mm] | 538.6 | 516.8 | **47.9** |
| | J2J [mm] | 630.1 | 583.5 | **42.2** |
| | Time [s] | 30.920 | **0.094** | 1.785 |
| Hi4D | V2V [mm] | 57.1 | 107.6 | **24.5** |
| | J2J [mm] | 66.6 | 123.7 | **20.6** |
| | Time [s] | 29.478 | **0.099** | 1.832 |
| PosePrior | V2V [mm] | 64.2 | **36.2** | 67.1 |
| | J2J [mm] | 75.9 | **42.3** | 75.7 |
| | Time [s] | 28.759 | 5.053 | **2.671** |

Table 1: Accuracy and runtime of SegFit on various datasets in comparison to state-of-the-art methods PTF and ArtEq.

## 4.2 ABLATION STUDY

To assess the contributions of individual components of our method, we conduct an ablation study and show the results in Table 2. We consider with the following variants:

**Without Body Parts**: We perform fitting without using body part segmentation, initializing the model with four different orientations to handle symmetry ambiguities.

**With Body Parts (No Centroid Initialization)**: We use body part segmentation but skip the centroid-based initialization step.

**Only Body Part Centroids**: We perform only the initialization step using body part centroids without further refinement.

As these results show, body part segmentation is crucial to the accuracy of SegFit, leading to a reduction in V2V error of 50% on BEHAVE and 966% on PosePrior, with a similar effect observed in the J2J error. Qualitative analysis indicates that this improvement is largely due to errors in the initialization of the pose. Furthermore, the complex poses in PosePrior are particularly challenging to model without part segmentation.

When evaluating SegFit with part segmentation but without the initialization step, the errors decrease significantly compared to those without segmentation, and are similar to those of the full method on BEHAVE, EgoBody, and Hi4D. However, the error remains higher across all datasets, with the V2V error increasing by 37% specifically on the PosePrior dataset, underscoring the importance of accurate initialization for complex poses.

|  | Metric | PosePrior | BEHAVE | EgoBody | Hi4D |
|---|---|---|---|---|---|
| w/o Body Parts |  | 383.9 | 55.6 | 109.7 | 83.6 |
| w/ Body Parts, w/o Centroid Init. | V2V [mm] | 91.9 | 39.9 | 48.5 | 27.5 |
| w/ Body Parts, only Centroid Init. |  | 135.8 | 97.3 | 105.2 | 93.0 |
| Full |  | **67.1** | **37.0** | **47.9** | **24.5** |
| w/o Body Parts |  | 362.8 | 43.6 | 100.3 | 71.0 |
| w/ Body Parts, w/o Centroid Init. | J2J [mm] | 98.3 | 32.6 | 42.7 | 22.8 |
| w/ Body Parts, only Centroid Init. |  | 122.5 | 80.8 | 87.9 | 75.9 |
| Full |  | **75.7** | **30.7** | **42.2** | **20.6** |
| w/o Body Parts |  | 12.699 | 6.051 | 4.281 | 5.842 |
| w/ Body Parts, w/o Centroid Init. | Time [s] | 4.226 | 2.978 | 3.704 | 8.590 |
| w/ Body Parts, only Centroid Init. |  | **0.408** | **0.404** | **0.400** | **0.403** |
| Full |  | 2.671 | 1.860 | 1.785 | 1.832 |

Table 2: Ablation study of SegFit variants on the PosePrior, BEHAVE, EgoBody, and Hi4D datasets.

Finally, we assess the accuracy of the initialization step on its own, which is approximately three times lower than that of the full method, demonstrating the impact of the refinement step. Despite the lower accuracy of this variant, it is worth noting that the accuracy remains fairly consistent across all four datasets, and the fitting time is between four and seven times lower than that of the full method, averaging below half a second. This presents an opportunity for a trade-off between accuracy and time efficiency, making it a suitable alternative in online environments.

## 4.3 IMPROVEMENTS TO BODY PART SEGMENTATION

After fitting the SMPL-X models, we reassign body part labels to the point clouds based on nearest-neighbor correspondences with the model vertices, resulting in improved segmentations. We compare the segmentation performance of Human3D before and after applying our method. Table 3 presents the accuracy, Intersection over Union (IoU), and Average Precision (AP) for each dataset.

| Method | Metric | PosePrior | Hi4D | BEHAVE | EgoBody |
|---|---|---|---|---|---|
| Human3D | Accuracy | **93.78**% | **96.50**% | 74.61% | 77.06% |
| + SegFit |  | 92.81% | 92.54% | **91.22**% | **84.71**% |
| Human3D | IoU | 75.26% | **84.04**% | 57.38% | 62.89% |
| + SegFit |  | **75.31**% | 75.67% | **77.26**% | **69.48**% |
| Human3D | AP | 82.34% | **91.89**% | 73.42% | 73.98% |
| + SegFit |  | **83.58**% | 85.88% | **85.74**% | **77.20**% |

Table 3: Comparison of body part segmentation performance before and after applying SegFit. We report the segmentation accuracy, Intersection over Union (IoU), and Average Precision (AP) for each dataset.

The results show significant improvements in segmentation accuracy on the BEHAVE and EgoBody datasets after applying SegFit, with accuracy increases of approximately 17% and 8%, respectively. This demonstrates that our method effectively refines the segmentation by leveraging the fitted models. Both Hi4D and PosePrior are very clean datasets with high-quality point clouds and barely any occlusions. Thus, Human3D has already achieved high accuracy, which we cannot improve. This suggests that our method may not provide further improvements when the initial segmentation is already highly accurate. However, this is not the case for unseen real-world datasets like BEHAVE and EgoBody.

### 4.3.1 FINE-TUNING OF HUMAN3D

To evaluate the potential of our method to enhance segmentation, we fine-tune Human3D using the improved segmentations from SegFit as pseudo-ground truths. We focus on the BEHAVE and EgoBody datasets, where the initial segmentation accuracy of Human3D is lower. Table 4 presents the performance of Human3D before and after fine-tuning on BEHAVE and EgoBody.

| Human3D Model | Metric | BEHAVE | EgoBody |
|---|---|---|---|
| w/o Fine-Tuning | Accuracy | 74.61% | 77.06% |
| w/ Fine-Tuning | | **89.96%** | **88.80%** |
| w/o Fine-Tuning | IoU | 57.38% | 62.89% |
| w/ Fine-Tuning | | **75.36%** | **78.88%** |
| w/o Fine-Tuning | AP | 73.42% | 73.98% |
| w/ Fine-Tuning | | **84.56%** | **84.61%** |

Table 4: Segmentation performance of Human3D (Takmaz et al., 2023) before and after fine-tuning in a self-supervised manner on the outputs of the proposed SegFit.

Fine-tuning Human3D on BEHAVE leads to a substantial increase in accuracy, from 74.61% to 89.96%, and similar improvements in IoU and AP metrics. This indicates that our method effectively generates high-quality pseudo-ground truths that can be used to enhance segmentation networks on new datasets without manual annotations. On EgoBody, which was part of Human3D's original training data, we still observe improvements after fine-tuning, suggesting that our method can further enhance performance.

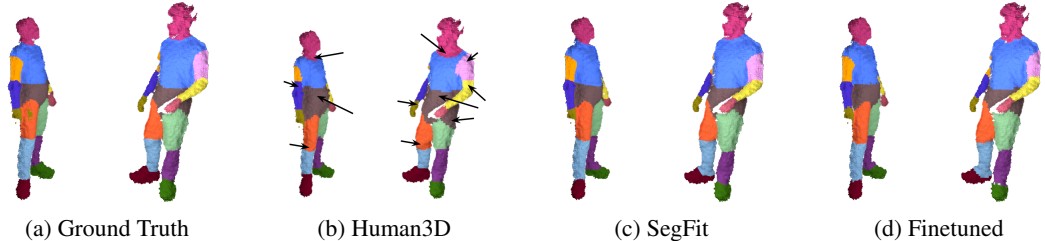

(a) Ground Truth      (b) Human3D      (c) SegFit      (d) Finetuned

Figure 3: Body part segmentation of a scan from the EgoBody dataset, as predicted by Human3D (Takmaz et al., 2023) before finetuning (3b), our proposed method SegFit (3c), and Human3D after finetuning on the pseudo ground truths provided by SegFit (3d). The arrows in (3b) highlight how Human3D's separation of body parts was originally imprecise and was corrected by SegFit.

## 5 CONCLUSION

In this paper, we introduce SegFit, a novel hybrid approach for fitting parametric human body models to diverse 3D point clouds. SegFit combines body part segmentation, based on Human3D (Takmaz et al., 2023), with SMPL-X modeling (Pavlakos et al., 2019) to iteratively enhance both segmentation and pose fitting accuracy. Our method demonstrates significant improvements over state-of-the-art techniques such as PTF (Wang et al., 2021b) and ArtEq (Feng et al., 2023b), especially in complex, real-world scenarios where occlusions, human-object interactions, and multiple human instances are prevalent.

By incorporating an initial body part segmentation and refining it through SMPL-X model fitting, SegFit creates a feedback loop where better pose fitting leads to improved segmentation and vice versa. This self-supervised cycle enhances the robustness and generalization capabilities of the method across diverse datasets, as shown by our experiments on the PosePrior (Zhang et al., 2021), BEHAVE (Bhatnagar et al., 2022), EgoBody (Zhang et al., 2022), and Hi4D (Yin et al., 2023) datasets. Our results indicate up to a tenfold improvement in pose estimation and a 15% increase in segmentation accuracy. Furthermore, the segmentation fine-tuning process with pseudo ground truths provides a versatile solution for unlabeled data, paving the way for more accurate self-supervised human modeling.

Our work sets a new benchmark for body fitting on point clouds, with the potential to significantly advance human representation in complex environments. Future work will explore potential improvements to the pose fitting accuracy, such as by introducing a penetration loss term for scenes where humans interact with each other or with objects in their environment.

## 5.1 REPRODUCIBILITY STATEMENT

We make the source code of our method available in a public repository. For finetuning Human3D using the outputs of our method, we refer the reader to Human3D's repository. All datasets used in this study have been used without alterations. The only exception to this is the separation of human instances in EgoBody and Hi4D before evaluating ArtEq and PTF as described in Section 4.1.

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
