# OpenReview forum: "SegFit: Robust SMPL-X Fitting with Body Part Segmentation on Real-World Point Clouds"
_ICLR.cc/2025/Conference — ICLR 2025 Conference Withdrawn Submission_

### Official Review · Reviewer_n7RW · 2024-10-15

**Soundness:** 2
**Presentation:** 2
**Contribution:** 2
**Rating:** 3
**Confidence:** 4

**Summary:**

The authors present a domain-adaptation method for human body-part segmentation from 3D point clouds. They register a parametric body model to scans to produce pseudo-ground-truth exemplars which are then used to finetune the segmentation model. Estimated segmentations are incorporated into the point-cloud registration during initialization. The authors demonstrate that the segmentation-bootstrapping method improves generalization performance on some datasets. They also evaluate the usefulness of the estimated segmentations for registration initialization.

**Strengths:**

1. The method is intuitive and fairly easy to understand.
2. The text is generally well-written.
3. The pseudo-ground-truth seems to improve segmentation performance on some datasets.

**Weaknesses:**

1. The paper is greatly lacking in experimental detail which makes it difficult to evaluate. It's not clear even what data the models were trained on. See `Questions` for elaboration. It is imperative that the authors clarify whether the segmentation model and the baseline methods were trained on the same data. If not, the "robustness" claims appear somewhat dubious.
2. The authors claim "This enhanced segmentation serves as pseudo ground truth to fine-tune the segmentation network in a self-supervised manner, creating a feedback loop where improvements in pose fitting lead to better segmentation and *vice versa*." (emphasis added) but appear to only evaluate the effect of the finetuning on segmentation performance, leaving the pose-fitting portion of the claim seemingly unsubstantiated. Relatedly, the authors should consider changing the paper title as the focus of the work appears to be segmentation, with point-cloud registration appearing primarily as a means.
3. In the conclusion, the authors write "This self-supervised cycle enhances the robustness and generalization capabilities of the method across diverse datasets, as shown by our experiments on the PosePrior (Zhang et al., 2021), BEHAVE (Bhatnagar et al., 2022), EgoBody (Zhang et al., 2022), and Hi4D (Yin et al., 2023) datasets," highlighting generalization of the segmentation method to Hi4D. However, it must be pointed out that the adaptation method failed on Hi4D, requiring supervised finetuning: "Furthermore, as Human3D struggles to differentiate between touching instances in Hi4D, we finetune on this dataset before evaluating SegFit." The authors seem to forget about this and later attribute high performance on Hi4D to data quality: "Both Hi4D and PosePrior are very clean datasets with high-quality point clouds and barely any occlusions. Thus, Human3D has already achieved high accuracy, which we cannot improve."
4. The authors have demonstrated improved performance through finetuning the segmentation model on pseudo-ground-truth samples, but do not seem to have evaluated whether the nearest-neighbor segmentations obtained through the registration are more accurate: "Use the fitted SMPL-X meshes to reassign part labels to the point cloud via nearest-neighbor matching, resulting in enhanced segmentation."
5. The authors' reference to prior work is sloppy and the papers cited often do not match the sentences in which they appear. This includes all citations in the motivation of the work:
> Fitting a parametric human body model to 3D point cloud data (measured by, e.g., Lidar or Kinect) without contextual guidance can be challenging and often yields imprecise results (Bogo et al., 2016; Kolotouros et al., 2019). However, having prior information about body parts can significantly improve this process (Varol et al., 2017; Zanfir et al., 2018; Xu et al., 2020). Thus, this paper focuses on incorporating body part segmentation into the body fitting process.

    The first four referenced works, SMPLify, SPIN, SURREAL, and Zanfir et al. 2018, are image-based HPS methods and do not deal with 3D point-cloud data. GHUM involved point-cloud registration, but it's not clear how it relates to the claim. This continues throughout the paper. As a second example, from the following quote it might be understood that the cited works are quantitatively compared against:
    > Our experiments demonstrate significant improvements in both pose modeling accuracy and body part segmentation performance compared to leading methods (Kolotouros et al., 2019; Kocabas et al., 2020; Zanfir et al., 2018), especially in the complicated scenes.

    However, these are image-based HPS works, none of which are compared against nor even could they be.

**Questions:**

1. What data were the baselines trained on for each experiment?
2. What data were Human3D trained on for each experiment?
3. The authors write they "Fine-tune the Human3D segmentation network using the improved segmentation as pseudo ground truth, enhancing its performance on new data." but don't specify what data samples are used for this finetuning. Are the evaluation samples actually "new" or do they overlap with the samples that pseudo-ground-truth segmentations are bootstrapped from? If there is overlap, this must be made clear and included in the time calculation.
4. How many iterations is the "self-supervised cycle" applied for?
5. It is not clear how the segmentations are employed during model fitting. The optimization objective does not seem to contain a segmentation term. Is it just used during initialization and for nothing else? Are segmentations used to filter out points? That there are four options in Table 2 is confusing. The differences between these must be made clear.
6. Which version of VPoser was employed and what was it trained on? "To further enforce realistic poses and avoid unnatural articulations, we utilize VPoser, a variational human pose prior trained on a large dataset of human poses." Overlap with PosePrior, "A subset of the AMASS dataset containing complex human poses with limbs extended at extreme angles, posing difficulties for accurate pose," would complicate evaluation of generalization.

Additional Comments:
1. The authors write "We make the source code of our method available in a public repository." but do not seem to have submitted supplementary materials.
2. The bibliography is not consistently formatted and includes several duplicate entries.
3. The authors should have evaluated against LoopReg as its main contribution is self-supervised fitting.

---

### Official Review · Reviewer_UavD · 2024-11-01

**Soundness:** 3
**Presentation:** 4
**Contribution:** 2
**Rating:** 5
**Confidence:** 4

**Summary:**

The paper proposes a hybrid approach to fit a parametric model of the human body to point clouds. The approach first segments the point cloud into body parts and, from these body parts, computes an initial, plausible alignment of the SMPL-X template model to the point cloud observation. Next, the approach proposes to optimize the model parameters by minimizing the point to vertex distance between the point cloud and the parametric model. The fitted model is then used to refine the segmentation and fine-tune the model on datasets, without requiring ground truth annotations.

**Strengths:**

The presentation of the paper is very clear; it is well-written, and easy to follow.
The evaluation uses a good diversity of datasets to showcase the strengths of the proposed approach. The proposed algorithm also performs better on three of the four datasets compared to the baselines and the different components are properly ablated.
Adding the application of self-supervised fine-tuning of a body part segmentation network also adds an interesting application, which can be useful to the community.

**Weaknesses:**

I summarize my questions and concerns here:
1. Why does the paper fixate so much on relying only on the point clouds for model fitting? All the datasets used in the paper contain RGBD data, so it would be natural to integrate the RGB information into the model. Also, as far as I know, the current SOTA of SMPL model fitting relies largely on multi-view RGB information.
2. Line 279. How is VPoser used in the optimization? These details seem to be missing from the paper and should be added.
3. Section 3.6: What is the reasoning behind using a direct assignment of the segmentation using the nearest point on the SMPL model? Would a probabilistic/voting scheme that considers more neighbors and the current assignment of the point not be more robust?
4. Line 305: "By training the network on these improved labels, we enable it to generalize better to new, unlabeled data and enhance its segmentation accuracy in diverse scenarios" -> This is untrue. The paper proposes to use data of a specific dataset and fine-tune the segmentation model on this specific data. This does not naturally lead to better generalization performance, as the segmentation model will be adapted to perform well on this one specific dataset. I suggest changing this sentence to reflect this or adding an experiment that confirms that fine-tuning a model on a single dataset leads to better generalization. If this statement were true, we could finetune the model on the BEHAVE dataset and apply it on the EgoBody dataset to achieve better performance and vice versa.
5. Table 1. Why does ArtEq perform better on the PosePrior dataset? On PosePrior, ArtEq also takes much longer to optimize than on the other datasets. Does this mean there exists an error on the other evaluations? In contrast, the proposed methods and PTF take always about the same time on all datasets. It seems suspicious that ArtEq does so much better on one dataset but not on the others. Please explain the difference.
6. How would the proposed approach compare to Multi-view or even single-view methods that use RGB-D data instead of only point clouds?
7. Table 3. The fact that Human3D does better in some instances than when we do reassignment based on simple nearest neighbor seems to indicate that a probabilistic approach that considers multiple factors would be better.

**Questions:**

See above. Overall, I think the paper is well presented and motivated, but some gaps must be addressed. As it stands, I would recommend rejecting the paper, but I will consider the rebuttal to my questions in my final score.

---

### Official Review · Reviewer_WCUi · 2024-11-03

**Soundness:** 3
**Presentation:** 3
**Contribution:** 2
**Rating:** 5
**Confidence:** 4

**Summary:**

This paper presents a method for segmenting human body parts from real-world point clouds. Instead of concentrating on human body segmentation, this paper co-optimizes human pose estimation and human body segmentation.

**Strengths:**

The concept of co-optimizing body pose and human point cloud segmentation is both intriguing and logical. The structure of the paper is well-organized, and Section 3.2 effectively illustrates the central idea.

**Weaknesses:**

The innovation of this paper is somewhat lacking. While the main contribution is in unsupervised body part segmentation, it appears that a good initialization is crucial for subsequent processes. However, this initialization relies on a supervised method called "3D Segmentation of Humans in Point Clouds with Synthetic Data."

Additionally, the paper lacks essential experimental analysis. Table 1 indicates significant performance improvements on the "BEHAVE" and "EgoBody" datasets, but the enhancement on the "PosePrior" dataset is limited.

**Questions:**

Please refer weakness.

---

### Official Review · Reviewer_Xy8J · 2024-11-03

**Soundness:** 2
**Presentation:** 2
**Contribution:** 2
**Rating:** 5
**Confidence:** 5

**Summary:**

This paper propose a method to fit parametric human body models to noisy 3D point cloud. The proposed method, SegFit, first segments the point cloud into body parts using existing method -- Human3D, followed by a series of carefully-crafted optimization & finetuning stages to fit SMPL-X template to the segmented point cloud. Despite simple, the evaluation suggests state-of-the-art results.

**Strengths:**

Fitting to 3D point clouds, or sometimes called registrations, has long been studied in 3D Vision/Graphics community. Besides the observed 3D points, recent state-of-the-art methods often leverage 2D detectors, e.g., human joint detection as a complementary supervision signal. This is how existing benchmarks built their ground truths, e.g., BEHAVE, EgoBody, InterCap, AGORA. This submission, on the other hand, is completely 3D-based; there is no 2D joint nor part detection involved at all, and yet the reported J2J errors are in a similar range with state-of-the-art 3D pose estimators.

**Weaknesses:**

The components are all very simple. I've been working on human motion tracking and pose estimation for more than a decade, either fitting to 2D and/or 3D data. I know very well how brittle a SMPL-based fitting method is and how sensitive it is to the noise in observations. The proposed method consists of only a data term and a few standard prior terms, e.g., Eq. 5, which is very easy to break.
The 3D observations can contain missing data, e.g., Fig. 1(a), and/or outliers, backpacks, and loose apparel, and yet there is even no temporal smoothness term in the proposed method. I am missing some key insight why it works well, so I dropped a few concrete questions below. While I put "marginally below the acceptance threshold," it is really borderline. If the questions get addressed with evidence, I am happy to increase the rating.

**Questions:**

1. Sensitivity to errors in segmentation: point clouds can contain missing data and outliers. Please provide some sensitive analysis on the robustness of fitting to wrong segmentation, e.g., perturb the segmentation results with varied noise level and see how the fitting accuracy changes accordingly.
2. More visuals: (1) segmentation results of Fig. 1(a), (2) model initialization of Fig. 1 (after Sec. 3.4 before Sec. 3.5), (3) qualitative results when point clouds contain outliers, e.g., backpacks, objects, and loose apparel. (4) qualitative results when point clouds contain significant missing data, e.g., missing arms and legs caused by occlusion.
3. Clear breakdown of timing reported in Table 1. Does 1.86s (BEHAVE) or 1.785s (EgoBody) mean the timing for the whole pipeline? That is, from segmentation, model initialisation, to self-supervised fine-tuning. If some parts are ignored for fair comparison, please state it clearly.
4. How many iterations of fitting <-> fine-tuning are performed in the experiments? Is it a fixed hyperparameter, or is there some other stopping criteria?

---

### Note · Authors · 2024-11-15

I have read and agree with the venue's withdrawal policy on behalf of myself and my co-authors.